# Learning to Communicate Locally for Large-Scale Multi-Agent Pathfinding

## Abstract

Multi-agent pathfinding (MAPF) is a widely used abstraction for multi-robot trajectory planning problems, where multiple homogeneous agents move simultaneously within a shared environment. Although solving MAPF optimally is NP-hard, scalable and efficient solvers are critical for real-world applications such as logistics and search-and-rescue. To this end, the research community has proposed various decentralized suboptimal MAPF solvers that leverage machine learning. Such methods frame MAPF (from a single agent perspective) as Dec-POMDP when at each time step an agent has to decide an action based on the local observation and typically solve the problem via reinforcement learning or imitation learning. We follow the same approach but additionally introduce a learnable communication module tailored to increase the level of cooperation between the agents via efficient feature sharing. We present the Local Communication for Multi-agent Pathfinding (LC-MAPF), a foundation model that applies multi-round communication between neighboring agents to exchange information and improve their coordination. Our experiments show that the introduced method outperforms the existing learning-based MAPF solvers, including IL and RL based approaches, across diverse metrics in a diverse range of (unseen) test scenarios. Remarkably, the introduced communication mechanism does not compromise the scalability LC-MAPF, which is a common bottleneck for communication-based MAPF solvers.

## 1 Introduction

Modern robotic systems often involve multiple mobile agents that must navigate and operate within shared environments, such as robots transporting goods in automated warehouses (Li et al., 2021a) or autonomous vehicles on public roads (Li et al., 2023). A key abstraction for modeling and solving the challenge of coordinating such agents safely is multi-agent pathfinding (MAPF) (Stern et al., 2019).

In MAPF, time is divided into discrete steps, and agents move on a graph structure (typically a 4-connected grid). Each agent acts synchronously, with each action, either moving to a neighboring vertex or waiting in place, taking exactly one time step. The goal is to compute a set of individual plans, one for each agent, that ensures no collisions occur as the agents execute them.

Many challenges of real-world robotics are not directly captured by the MAPF abstraction, including continuous space and time, asynchronous agent behavior, limited communication and observation, and various perception constraints. Despite these simplifications, MAPF successfully models the central difficulty in multi-robot navigation: coordinating agents to avoid collisions while aiming to optimize a specific cost function. As a result, MAPF has attracted substantial interest from both the robotics and AI research communities. Furthermore, a number of studies have demonstrated the successful application of MAPF-based methods to the continuous, noisy, and uncertain environments faced by real-world robotic systems (Hönig et al., 2016; Ma et al., 2019a).

MAPF is most commonly approached in a centralized setting, where a single planner with full knowledge of the environment is responsible for generating plans for all agents. A wide range of both optimal and suboptimal centralized solvers have been proposed (Standley, 2010; Sharon et al., 2015; Wagner & Choset, 2011; Surynek et al., 2016; Okumura et al., 2022; Okumura, 2023; Li et al., 2022; Veerapaneni et al., 2024; Wang et al., 2025).

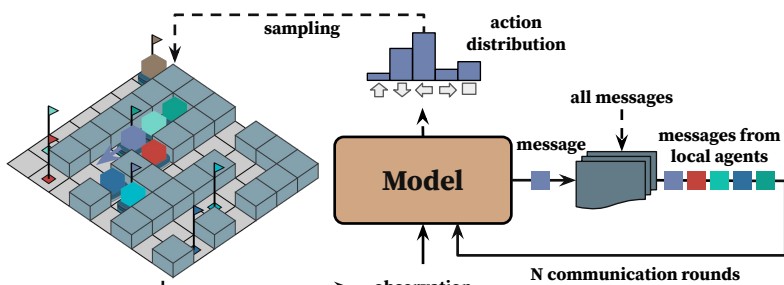

Figure 1: Overview of LC-MAPF communication process. At each step, the environment provides local observations to all agents. Based on the observation, each agent generates a message and exchanges it with neighboring agents over multiple communication rounds. After the discussion phase, the agent selects an action based on the aggregated information. This iterative process enables decentralized coordination through learned communication.

It is well established that optimal MAPF solvers scale poorly with the increasing numbers of agents, as the problem is NP-hard (Surynek, 2010). Suboptimal solvers, on the other hand, can scale to thousands of agents, but their solution quality may degrade significantly. Consequently, a central focus of MAPF research is striking the balance between computational efficiency and solution quality.

One promising strategy for addressing this challenge is to adopt a decentralized approach. Here, MAPF is modeled as a decentralized sequential decision-making problem, where each agent independently selects and executes actions at every time step based on local observations. The resulting decision-making policy may be fully learned or designed as a hybrid, combining learnable and fixed components (Liu et al., 2020; Li et al., 2021b; Wang et al., 2023; Ma et al., 2021a;b; Tang et al., 2024; Skrynnik et al., 2024; 2023; Sagirova et al., 2025; Phan et al., 2025). A recent survey provides a comprehensive overview of developments in this area (Alkazzi & Okumura, 2024).

One of the recent advancements in decentralized, learnable MAPF is MAPF-GPT (Andreychuk et al., 2025), which relies entirely on supervised learning using a transformer-based neural network trained on an extensive dataset of approximately one billion observation-action pairs. Despite its simplicity, MAPF-GPT outperforms most of the state-of-the-art learning-based MAPF methods.

However, a major limitation of MAPF-GPT is its lack of agent-to-agent communication. While it learns collaborative behavior through the data, it does so without any communication between agents, as the training data is generated by a centralized solver that does not include communication signals. This means that MAPF-GPT cannot explicitly facilitate interaction or collaboration between agents during problem-solving, which could be a key factor in improving performance.

Several existing decentralized MAPF methods, such as SCRIMP, PICO, DCC, and DHC use communication mechanism. However, it is mostly limited to sharing local observations or internally known state information in one round of communication (Alkazzi & Okumura, 2024). These mechanisms often fall short of enabling agents to engage in more meaningful coordination.

We argue that effective communication in decentralized MAPF should extend beyond single-message exchange and involve multiple rounds of agent interaction. Such iterative communication enables agents to negotiate, resolve conflicts, and build consistent joint plans that are crucial for robust multi-agent coordination in complex environments. Motivated by this, we explore how to equip a large transformer-based imitation learning model with the ability to perform effective local communication.

Our main contributions are the following:

- We introduce a novel communication learning framework (see Figure 1 for an overview) called **LC-MAPF**[1], which enables agents to communicate using only the expert demonstrations of selected actions, without requiring explicit communication supervision.

---

[1]Source code: https://anonymous.4open.science/r/LC-MAPF-18734

- We present a transformer-based model with 2 million parameters that significantly improves performance and sets a new state-of-the-art among learnable decentralized MAPF solvers. We conduct extensive evaluations and compare it with existing learning-based approaches.

- Additionally, we extensively study how the number of communication rounds influences the performance of the agents, as shown in the ablation study. Moreover, we show that despite incorporating communication, the proposed mechanism maintains linear scalability as the number of agents grows.

## 2 RELATED WORK

The related work section covers three categories relevant to the proposed approach: foundation models for multi-agent systems, communication-based learning in MAPF, and multi-agent pathfinding.

### 2.1 FOUNDATION MODELS FOR MULTI-AGENT SYSTEMS

Foundation models are typically trained on large-scale datasets, enabling generalization through zero-shot or few-shot learning (Bommasani et al., 2021; Yang et al., 2023). For autonomous agents, demonstrations of task execution in the environment are used as a dataset, and generalization implies the execution of new tasks that were not in the training data distribution without additional demonstrations or with a minimal number of them (Firoozi et al., 2023). Demonstration-based pretraining is not commonly used in multi-agent settings, but there are some examples, including games such as chess (Silver et al., 2016; Ruoss et al., 2024), cooperative video games via self-play (Berner et al., 2019), and multi-agent pathfinding, as in SCRIMP (Wang et al., 2023).

A key strength of foundation models is their fine-tuning capability, which supports rapid adaptation to task-specific requirements. While widely adopted in robotics, particularly in multimodal tasks involving text-based instructions (Firoozi et al., 2023; Team et al., 2024; Kim et al., 2024), their use in multi-agent systems remains relatively limited. Notable examples include the Magnetic-One model for language and multimodal tasks in WebArena (Fourney et al., 2024) and MAPF-GPT for decentralized pathfinding (Andreychuk et al., 2025).

### 2.2 MULTI-AGENT PATHFINDING

A variety of approaches have been proposed for solving MAPF. Rule-based solvers are designed for fast computation but lack guarantees on solution quality (Okumura, 2023; Li et al., 2022). Reduction-based methods convert MAPF into classical problems such as minimum-cost flow or SAT, leveraging existing solvers to compute optimal solutions (Surynek et al., 2016). Search-based solvers, such as CBS and its variants (Sharon et al., 2015; 2013; Wagner & Choset, 2011), apply graph search techniques and often offer optimality or bounded-suboptimality guarantees. Simpler methods like prioritized planning (Ma et al., 2019b) trade off optimality for efficiency and scalability.

### 2.3 COMMUNICATION-BASED MAPF METHODS

More recently, learning-based approaches have emerged. PRIMAL (Sartoretti et al., 2019) was among the first to demonstrate decentralized MAPF solving via learning. In case of PRIMAL the only communication between agents is their corresponding targets. One of the first learnable MAPF solvers that has a specific learnable communication block was DHC (Ma et al., 2021a) that demonstrate significant improvement over PRIMAL. Subsequent methods such as DCC (Ma et al., 2021b) utilize the ideas proposed by DHC, but enhance the communication mechanism by training selective communication. Another approach, SCRIMP (Wang et al., 2023), combines imitation learning, reinforcement learning and communication mechanism and improves the efficiency even further. Another example of a decentralized communication approach coming from the online MAPF is the SRMT (Sagirova et al., 2025). It allows agents to implicitly exchange information by generating and broadcasting agents' working memory representations learned by the memory-augmented transformer (Burtsev et al., 2020). The memory states used for communication, are updated recurrently (Bulatov et al., 2022) to preserve the historical information and improve agents coordination.

## 3 BACKGROUND

### 3.1 PROBLEM STATEMENT

MAPF problem is a tuple $(G, s^1, ..., s^n, g^1, ..., g^n)$, where $G = (V, E)$ is a graph representing the environment, $s^i \in V$ is the start vertex of the $i$-th agent, and $g^i \in V$ is its goal vertex. Totally $n$ agents ($\mathcal{A} = \{u_1, \ldots, u_n\}$) are presented in the environment. The task is to find a set of plans $Pl = \{pl^i\}$ composed of the actions that can be either move to an adjacent vertex or stay at the current vertex. Additionally, the plans should be conflict-free, i.e., no two agents occupy the same vertex or traverse the same edge at the same time. The solution cost is typically measured by either the *Sum-of-Costs*, $SOC(Pl) = \sum_{i=1}^{n} cost(pl^i)$, or the *makespan*, $msn(Pl) = \max_{i=1}^{n} cost(pl^i)$, where $cost(pl^i)$ is the timestep at which agent $i$ reaches its goal and remains there.

MAPF can also be formulated as a sequential decision-making problem, where the task is to construct a centralized policy $\pi_{\text{centralized}}$ that selects a joint, conflict-free action $\mathbf{a} = a^1 \times \cdots \times a^n$ at each timestep, with $a^i$ denoting agent $i$'s action. Such a policy can be hand-crafted or learned.

Finally, MAPF can also be treated as a decentralized decision-making problem where the goal is to learn a homogeneous individual policy $\pi$ shared by all agents, which selects an action for each agent based solely on local observations and, possibly, communication. The observations typically include information about nearby obstacles and agents, rather than the full global state.

### 3.2 IMITATION LEARNING FOR MAPF

Imitation learning seeks to approximate an expert policy $\hat{\pi}$ by training a parameterized policy $\pi_\theta$. A dataset $\mathcal{T}$ of expert trajectories is first collected: $\hat{\mathcal{T}} = \{\hat{\tau}_i\}_{i=1}^{K}$, where each trajectory $\hat{\tau}_i = \{(s^1, \mathbf{a}^1), \ldots, (s^L, \mathbf{a}^L)\}$ of length $L$ consists of state and joint action pairs. In MAPF, $\hat{\pi}$ is typically a centralized solver, for example, LaCAM* (Okumura, 2024).

To enable decentralized learning, individual agent trajectories $\tau_u^{\hat{\pi}} = \{(o_u^1, a_u^1), \ldots, (o_u^L, a_u^L)\}$ are extracted, where $o_u^t$ is the local observation of agent $u$ at time $t$, and $a_u^t$ is the corresponding expert action. Observations may be represented as tensors or token sequences (e.g., in transformer-based models (Ruoss et al., 2025)). The resulting dataset $\mathcal{D} = \{\tau_u^{\hat{\pi}}\}_{u=1}^{n}$ is then used to train the policy.

The learning objective minimizes the negative log-likelihood of expert actions:

$$\theta^\star = \arg \min_\theta \mathbb{E}_{(o_u, a_u^{\hat{\pi}}) \sim \mathcal{D}} \left[ -\log \pi_\theta(a_u^{\hat{\pi}} \mid o_u) \right]. \tag{1}$$

After training, actions are sampled as $a^u \sim \pi_\theta(o_u)$.

## 4 METHOD

### 4.1 LOCAL COMMUNICATION MAPF

The scheme for the proposed communication workflow is presented in Figure 2. At each time step $t \in [1, \ldots, L]$ for each agent $u \in [1, \ldots, U]$, the models takes as input an observation $o_u^t$

$$o_u^t = [\text{cost-to-go}_u^t, i_u^t, n_{u,1}^t, \ldots, n_{u,k}^t], \tag{2}$$

presented by the tokenized sequence of egocentric cost-to-go matrix cost-to-go$_u^t$, information $i_u^t$ about the agent $u$ and its $k$ local neighboring agents $n1_u^t, \ldots, n_k{}_u^t$. Information about agents contains relative coordinates of current and goal locations, action history for previous $k$ steps, and a greedy action. The model also takes as input a communication round chat $c_u^t$:

$$c_u^t = [m_i^t, m_{n_1}^t, \ldots, m_{n_k}^t], \tag{3}$$

where $m_i^t$ refers to the agent $u$ message and $m_{n_1}^t, \ldots, m_{n_k}^t$ are the neighboring agents messages. The communication is presented by the cycle of several consecutive rounds of message generation and exchange, resulting in the prediction of the actions used for training on expert data. This type of multi-round communication is conceptually similar to message passing in graph neural networks. In our case, though, each round is implemented by a multi-layer Transformer that jointly attends to dynamically selected local neighbors and the agent's

egocentric observation history. The backbone model used for data processing is a transformer (GPT) with non-causal attention mask, with linear layers for message and action prediction heads.

Each agent generates its message based on the information about itself and the nearby ones. Considering that agents with adjacent locations independently use information about each other to generate their messages, we augment $o_u^t$ with positional encoding and embedded representations of global agent identifiers $id_u$:

$$\tilde{o}_u^t = o_u^t + \text{PosEnc}(o_u^t) + \text{Emb}_\text{o}(id_u). \quad (4)$$

In the same way, we enrich chat representations $c_u^t$ with the specified agent identifiers embeddings to specify to which agent belongs each message:

$$\tilde{c}_u^t = c_u^t + \text{Emb}_\text{c}(id_u). \quad (5)$$

To create such identifiers for a system of $L$ agents interacting in the environment, we sample $L$ unique $h$-dimensional vectors randomly sampled from a uniform distribution over $[0, 1)$, where $h$ is the GPT hidden dimensionality. Such vectorized identifiers do not depend on the overall number of agents in the particular episode, providing a flexible instrument for unique 'naming' of agents' populations of various sizes, which is important for the method's scalability.

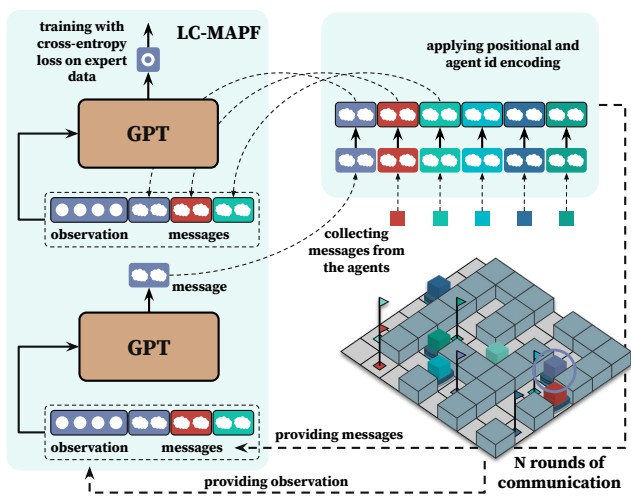

Figure 2: **Local Communication for MAPF approach.** At each communication round, agents firstly generate the current chat round messages based on the observation and the previous round chat, including messages from the considered agent and its neighbors. Secondly, the generated messages from all agents are used to update each individual agent chat state. Updated chat along with observation are used to predict the action and compute the cross-entropy loss for training. The chat update and action generation procedure repeats until the desired number of communication rounds is reached. The action distribution predicted in the final communication round is used to sample actions and apply them in the environment during the execution phase.

After all the preparations, we start the first round of communication. At start, agents have observations and no messages to exchange, so to initialize the communication cycle, we use a set of zero message vectors $c_{0u}^t$ modeling the empty chat history, and pass the concatenated inputs to the GPT:

$$g_u^t = \text{GPT}([\tilde{o}_u^t, \tilde{c}_{0u}^t]). \quad (6)$$

Next, we use the $i$-th element of $g_u^t$ corresponding to the considered agent message $m_i^t$ slot in the GPT output sequence to recurrently update it via the message generation head:

$$m_{1u}^t = \text{MsgHead}(g_u^t[i]). \quad (7)$$

As a result, we obtain the $m_{1u}^t$ - the current round message representation for the agent $u$. As soon as the messages for all $L$ agents are generated, we construct the first round chat $c_{1u}^t$ that will be processed by the GPT with action prediction head:

$$a_{1u}^t = \text{ActionHead}(\text{GPT}([\tilde{o}_u^t, \tilde{c}_{1u}^t])). \quad (8)$$

The predicted action is used to calculate the cross-entropy loss for this round. On the next round of communication, we will repeat the process described in Equations 6-8 using the same observation $o_u^t$ and the updated chat $c_{1u}^t$ as inputs. As a result, at each communication round, we use the observation and the historical information about agents' conversation from previous rounds to update the chat state with the current round of messages and make the decision about actions based on the actual information. Such sequential procedure of message generation followed by the action prediction instead of parallel generation of messages and actions after a single GPT forward pass allows to train the model without supervision on the message predictions using only the action prediction loss to update all the model weights including the message generation head.

After the given number of chat rounds, we aggregate the loss values to perform the optimization step and backpropagate the gradients. The end-to-end pathfinding objective optimization affects the message content and, consequently, the communication through: the parameters generating messages are updated only through backpropagation of the action prediction loss, with no explicit supervision on message content. The network learns to encode information most beneficial for coordination. Below, the model backward pass is provided to better illustrate the gradient flow that enables the network to learn effective communication.

At time step $t$ (we omit time step marker in formulas below for readability) and communication round $r \in [1, \ldots, R]$, an agent $u$ uses the previous round chat $\tilde{c}_u^{r-1}$ generates transformer output representation $g_u^r$ (Eq. 6). Then $g_u^r$ is used to update the message vector $m_u^r$ (Eq. 7). Finally, agents construct current round chats $\tilde{c}_u^r$ and concatenate them with observations $\tilde{o}_u$ to predict action logits $a_u^r$ (Eq. 8). The training objective is the summed cross-entropy loss:

$$\mathcal{L} = \sum_{r \in [1, \ldots, R]} \text{CE}\left(a_u^r, a_u^*\right) \tag{9}$$

During the backward pass message $m_u^r$ receives no direct supervision. However, it is included into the transformer input sequence for round $(r+1)$ for each agent in the system that considers agent $u$ as a neighbor. This is how each agent's message affects the action logits. The chain rule gives:

$$\frac{\partial \mathcal{L}}{\partial m_u^r} = \sum_{v \in \mathcal{N}_u} \sum_{\rho \in [r+1, \ldots, R]} \frac{\partial \mathcal{L}}{\partial a_v^\rho} \times \frac{\partial a_v^\rho}{\partial \tilde{c}_v^\rho} \times \frac{\partial c_v^\rho}{\partial m_u^{\rho-1}} \tag{10}$$

where $\mathcal{N}_u$ is the set of agents receiving $m_u^r$. Substitution of $m_u^r$ definition from Equation 7 gives:

$$\frac{\partial \mathcal{L}}{\partial g_u^r[i]} = \frac{\partial \mathcal{L}}{\partial m_u^r} \times \frac{\partial \text{MsgHead}(g_u^r[i])}{\partial g_u^r[i]}. \tag{11}$$

The gradients continue back into the GPT parameters, allowing the single shared loss $\mathcal{L}$ to sculpt each $m_u^r$ to carry exactly the information that reduces downstream action error, so agents learn the communication content purely via end-to-end backpropagation.

During the execution phase, the action generated at the last communication round is used to update the agent's state in the environment. For our experiments, we used 4 rounds of communication.

### 4.2 DATASET

To train LC-MAPF we have collected a new dataset with expert data. The way of collecting dataset mainly repeats the one made for MAPF-GPT. The dataset is collected on random and maze-like maps with size varying from 17x17 to 21x21 and 32 agents. The ratio between samples obtained on maze-like and random maps is 9:1, i.e. most of the data in the dataset is obtained from maze-like maps as they are more challenging than random maps. As an expert we utilized LaCAM* approach with 10 seconds timelimt. In contrast to the dataset of MAPF-GPT, that contains 1 billion samples, the collected dataset contains 32 million samples. The difference is explained by the fact that one sample in dataset of MAPF-GPT is an observation-action pair for a single agent, while in the dataset for LC-MAPF each sample contains observations and ground-truth actions for all 32 agents presented in the state. In addition, each sample in the collected dataset contains information about IDs of the observed agents to identify which of them need to communicate.

### 4.3 TRAINING AND TECHNICAL DETAILS

The training of LC-MAPF was performed on a server with 4xNVIDIA H100 GPUs for 100,000 iterations with batch size 30 (actual batch size is 960, as each sample has information about 32 agents) and 16 gradient accumulation steps. Thus, during training LC-MAPF has processed 1.5 billion single-agent observations within ˜95 hours. During training LC-MAPF performed 4 rounds of communication. The same number of rounds was used in experimental evaluation. The communication is possible only with agents, information about which is presented in the observation, which contains information about 13 nearest observable agents at most. Thus, each agent can receive up to 13 messages. More details and an explanation of the chosen limit are provided in the Appendix B.

Table 1 provides more details about hyperparameters used for training and the model.

## 5 Empirical Evaluation

### 5.1 Experimental Setup

The experimental evaluation was conducted on the POGEMA benchmark (Skrynnik et al., 2025) – a benchmark specifically designed to perform a comparison of learning-based MAPF solvers. It contains a variety of different types of maps – `Random`, `Mazes`, `Warehouse`, `Cities-tiles`, `Cities` and `Puzzles`. Each of them has its own specific which is able to demonstrate different aspects of the solvers, such as ability to efficiently coordinate hundreds of agents, to resolve complex collisions, to demonstrate generalization to unseen types of environments, etc. The benchmark also has an evaluation protocol and a set of high-level metrics, described below. *Performance* (*OOD*, *Cooperation*) is defined as $\frac{\text{SoC}_{\text{best}}}{\text{SoC}}$ if the instance is solved and 0 otherwise. *Scalability* is defined as $\frac{\text{runtime}(|\mathcal{A}_1|)}{\text{runtime}(|\mathcal{A}_2|)} \times \frac{|\mathcal{A}_2|}{|\mathcal{A}_1|}$. *Coordination* is defined as $1 - \frac{\text{Number of collisions}}{|\mathcal{A}| \times \text{episode length}}$. Finally, *Pathfinding* is defined as $\frac{\text{cost(best path)}}{\text{cost(found path)}}$ if a path is found and 0 otherwise.

| Parameter | Value |
|---|---|
| Minimum learning rate | 6e-5 |
| Maximum learning rate | 6e-4 |
| Learning rate decay | cosine |
| Warm-up iterations | 2000 |
| AdamW optimizer beta1 | 0.9 |
| AdamW optimizer beta2 | 0.95 |
| Gradient clipping | 1.0 |
| Weight decay | 1e-1 |
| Data type for computations | float16 |
| Gradient accumulation steps | 16 |
| Block size | 256 |
| Number of GPT layers | 5 |
| Number of attention heads | 5 |
| Hidden size | 160 |

Table 1: Values of hyperparameters used for training and the model of LC-MAPF.

Performance, Out-of-Distribution (OOD) and Cooperation metrics have the same formula but differ in the set of maps used to evaluate them. For Performance metric `Mazes` and `Random` maps are utilized, for OOD – `Warehouse` and `Cities-tiles`, while for Cooperation – `Puzzles`. Most of the metrics are relative and depends on the best-found solution cost ($SoC_{best}$). To save the consistency of the results with the ones presented in the benchmark, we have utilized the results of LaCAM* (Okumura, 2024) – centralized search-based solver, which shows best results among all the approaches presented in the benchmark. We have also utilized the results of evaluation of multiple state-of-the-art learning-based MAPF solvers that utilize communication – SCRIMP (Wang et al., 2023) and DCC (Ma et al., 2021b). Out of the other approaches, presented in the benchmark, we took the MAMBA approach (Egorov & Shpilman, 2022) as it shows the best results among all the presented pure MARL approaches. In addition to the proposed method, LC-MAPF, we have also evaluated MAPF-GPT. For evaluation of MAPF-GPT we utilized the trained model provided by its authors[2] with comparable size – 2 million parameters.

To obtain the results required to compute all metrics, each of the solvers is evaluated on 3376 different scenarios with up to 256 agents. In each run the episode length was set to the default values of POGEMA benchmark (128 in the most cases, except `Cities-tiles` – 256, and `Cities` – 2048). More details about number of instances, sizes of the maps, etc. can be found in (Skrynnik et al., 2025).

### 5.2 Experimental Results

The results of the main experiment are depicted in Figure 3. Despite LaCAM*, LC-MAPF substantially outperforms all baselines including MAPF-GPT in terms of Performance and Cooperation. In contrast to DCC and SCRIMP, whose communication mechanisms heavily depend on the total number of agents, LC-MAPF strictly considers only limited observable number of agents that allows scaling linearly to the number of agents and demonstrate perfect Scalability like its predecessor – MAPF-GPT. In terms of the rest metrics, such as Coordination, Pathfinding and Out-of-Distribution, LC-MAPF demonstrates comparable results.

All the approaches except MAMBA demonstrate high value of Coordination metric. However, it's worth noting that the number of collisions for SCRIMP is actually undefined as it has its own integrated environment and extra collision resolution technique that guarantees collision-free actions

---

[2]https://github.com/Cognitive-AI-Systems/MAPF-GPT

in the output. A scalability score of 1.0 for LC-MAPF, MAPF-GPT, and MAMBA indicates that the runtime grows proportionally with the number of agents, demonstrating that these learnable approaches scale linearly.

Noteworthy, among the communication-based learnable MAPF approaches (such as DCC and SCRIMP) only ours demonstrates linear scalability.

Further we investigate the influence of communication to the cooperative behavior of the agents and total number of collisions.

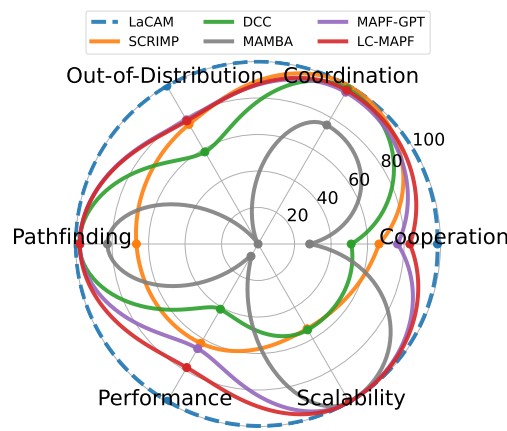

**Ablation study** During the ablation study of LC-MAPF we wanted to investigate the influence of the communication mechanism on the performance of the approach. To this end, we varied the number of communication rounds employed by LC-MAPF (from 1 to 6), as well as turned off the communication at all. The experiments were conducted on `Mazes` maps with number of agents varying from 8 to 64. For each number of agents all 128 testing instances provided by the POGEMA Behcnmark (Skrynnik et al., 2025) were used. The length of the episode was set to

Figure 3: Experimental results. Each metric is shown relative to LaCAM*, which is a fully-centralized MAPF solver that utilizes full knowledge of the environment.

128. Two performance indicators were tracked: success rate (the ratio of the successfully solved instances) and number of collisions. The results are shown in Table 2.

| | Success rate across different LC-MAPF communication rounds | | | | | | |
|---|---|---|---|---|---|---|---|
| Agents | Rounds=0 | Rounds=1 | Rounds=2 | Rounds=3 | Rounds=4 | Rounds=5 | Rounds=6 |
| 8 | 1.00 ± 0.00 | 1.00 ± 0.00 | 1.00 ± 0.00 | 1.00 ± 0.00 | 1.00 ± 0.00 | 1.00 ± 0.00 | 1.00 ± 0.00 |
| 16 | 0.98 ± 0.03 | 1.00 ± 0.00 | 1.00 ± 0.00 | 1.00 ± 0.00 | 1.00 ± 0.00 | 1.00 ± 0.00 | 1.00 ± 0.00 |
| 24 | 0.95 ± 0.04 | 1.00 ± 0.00 | 1.00 ± 0.00 | 1.00 ± 0.00 | 1.00 ± 0.00 | 1.00 ± 0.00 | 1.00 ± 0.00 |
| 32 | 0.81 ± 0.07 | 0.99 ± 0.01 | 1.00 ± 0.00 | 1.00 ± 0.00 | 1.00 ± 0.00 | 1.00 ± 0.00 | 1.00 ± 0.00 |
| 48 | 0.45 ± 0.09 | 0.87 ± 0.06 | 0.93 ± 0.04 | 0.92 ± 0.04 | 0.95 ± 0.04 | 0.97 ± 0.03 | 0.93 ± 0.04 |
| 64 | 0.20 ± 0.07 | 0.60 ± 0.09 | 0.69 ± 0.08 | 0.73 ± 0.07 | 0.72 ± 0.07 | 0.72 ± 0.08 | 0.70 ± 0.08 |
| | Collision counts across different LC-MAPF communication rounds | | | | | | |
| Agents | Rounds=0 | Rounds=1 | Rounds=2 | Rounds=3 | Rounds=4 | Rounds=5 | Rounds=6 |
| 8 | 4.8 ± 2.2 | 0.7 ± 0.3 | 0.6 ± 0.3 | 0.6 ± 0.2 | 0.5 ± 0.2 | 0.5 ± 0.2 | 0.5 ± 0.2 |
| 16 | 26.9 ± 5.8 | 5.1 ± 1.2 | 4.6 ± 1.0 | 3.9 ± 0.8 | 3.6 ± 0.7 | 3.7 ± 0.7 | 3.6 ± 0.7 |
| 24 | 75.4 ± 21.1 | 17.5 ± 3.6 | 13.1 ± 2.3 | 12.4 ± 2.1 | 12.0 ± 1.6 | 12.1 ± 1.8 | 13.2 ± 2.3 |
| 32 | 207.2 ± 54.7 | 42.0 ± 6.5 | 34.3 ± 5.4 | 33.1 ± 4.9 | 31.7 ± 4.1 | 34.2 ± 4.9 | 33.4 ± 5.5 |
| 48 | 805.2 ± 131.1 | 246.2 ± 53.0 | 193.2 ± 49.3 | 155.6 ± 24.0 | 141.2 ± 22.2 | 149.5 ± 24.1 | 152.0 ± 27.4 |
| 64 | 2186.7 ± 237.2 | 904.7 ± 149.7 | 669.4 ± 119.0 | 559.8 ± 90.8 | 562.5 ± 97.0 | 539.2 ± 105.5 | 536.6 ± 86.6 |

Table 2: Success rate and number of collisions of different versions of LC-MAPF and MAPF-GPT on `Mazes` map. The provided values are average ± 95% confidence interval. Tan boxes highlight the best mean values for visibility purposes.

Clearly the ratio of successfully solved instances as well as the amount of occurred collisions depends on the type of map and number of agents presenting in the scenario. The worst results, as expected, demonstrates LC-MAPF with disabled communication (rounds=0). The absence of any messages in the input leads to a heavy out-of-distribution for the model. As a result the model performs significantly worse that any other version. The rest LC-MAPF versions have communication but differ in the number of communication rounds. Looking at the results, it's evident, that a single round of communication is not enough and the performance of LC-MAPF can be enhanced by raising its number to at least 2. Further increase of number of communication rounds doesn't provide such evident profit in terms of success rate, but for sure reduces the number of collisions. The last fact indicates that communication mechanism trained by the model of LC-MAPF allows to choose more coordinated joint-actions with less collisions. As a part of the ablation study, in Appendix A and Appendix B we provide two additional experiments: the former demonstrates the

robustness of LC-MAPF to the communication errors, and the latter discusses the importance of the proposed communication bandwidth in terms of the communicating neighborhood size and the message vector size for the model performance.

**Scalability Analysis**   To better demonstrate the superior scalability of LC-MAPF, we present the actual decision times of all evaluated learning-based approaches with communication capabilities: DCC, SCRIMP, and LC-MAPF. These measurements were used to compute the Scalability metric. Table 3 shows the average time required for all agents to determine their next action across varying numbers of agents in the `Warehouse` map scenarios.

| Algorithm | 32 agents | 64 agents | 128 agents | 192 agents |
|-----------|-----------|-----------|------------|------------|
| DCC | $48.0 \pm 1.0$ | $164.0 \pm 2.0$ (×3.4) | $619.0 \pm 2.0$ (×12.9) | $1314.0 \pm 3.0$ (×27.4) |
| SCRIMP | $47.0 \pm 1.0$ | $106.0 \pm 1.0$ (×2.3) | $388.0 \pm 7.0$ (×8.3) | $1190.0 \pm 25.0$ (×25.3) |
| LC-MAPF | $117.0 \pm 3.0$ | $237.0 \pm 1.0$ (**×2.0**) | $462.0 \pm 2.0$ (**×3.9**) | $690.0 \pm 1.0$ (**×5.9**) |

Table 3: Decision time (in milliseconds) in the instances with different numbers of agents on `Warehouse` map.

Although LC-MAPF exhibits higher absolute values for small agent populations, its linear scaling properties become advantageous as complexity increases. When handling 192 agents, SCRIMP and DCC require 25.3 and 27.4 times more computation time, respectively, compared to their performance with 32 agents. The initially higher decision time of LC-MAPF in scenarios with fewer agents can be attributed to its multiple rounds of communication.

In practice, to mitigate the negative effect of LC-MAPF decision time in scenarios with smaller number of agents, the number of communication rounds can be decreased down to 2 rounds (or less, depending on the actual size of the considered agent population) without performance loss (as empirically shown in Table 2).

# 6   CONCLUSION

We introduced LC-MAPF, a novel communication learning framework for decentralized multi-agent pathfinding that leverages expert demonstrations without explicit communication supervision. The communication is organized in rounds to increase the level of cooperation between the agents. Our transformer-based model outperforms state-of-the-art learning-based MAPF solvers on the POGEMA benchmark, improving coordination and cooperation across diverse scenarios.

LC-MAPF maintains linear scalability with the number of agents, overcoming a common limitation of communication-based approaches. Ablation studies confirm that multi-round local communication enhances performance without sacrificing scalability or generalization. These results highlight LC-MAPF as a foundation model that offers an effective and scalable solution for decentralized multi-agent pathfinding through multi-round local communication.

## LIMITATIONS

The selection of agents for communication within the local field of view may be suboptimal, and alternative strategies for deciding communication participants could yield better results in certain scenarios. Another limitation is the use of a fixed number of communication rounds; ideally, this number should adapt to the complexity of the situation. For instance, when no other agents are nearby, communication might be unnecessary. While it can still benefit single-agent reasoning, this does not apply in the context of MAPF. Communication also introduces additional computational overhead, though it results in a more capable model. Finally, the method assumes homogeneous agents, not because of a methodological constraint but due to the specific MAPF formulation used. Supporting heterogeneous agents would likely require a more structured communication protocol, as communication in the latent space may no longer be feasible.

## REPRODUCIBILITY STATEMENT

Metrics are reported with 95% confidence intervals. All hyperparameters are specified in Table 1. We describe training details and used hardware in Section 4.3. We also release the full codebase to ensure reproducibility of results: `https://anonymous.4open.science/r/LC-MAPF-18734`.

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

APPENDIX

## A MESSAGE FAILURE TEST

There is a line of research where the communication in Dec-POMDPs is considered under various realistic circumstances, such as delay, failure, and cost (Wu et al., 2009; Lauri et al., 2019). In this section, we evaluate how LC-MAPF handles message transmission failures. For each agent, with the given probability, we replace the agent's updated message on each round with a random vector sampled from the standard normal distribution. We test the LC-MAPF with 20% and 50% message failure on a set of Random maps from the POGEMA benchmark and compared the results to the original model. The results are presented in the Table 4.

| | 0% failure | | 20% failure | | 50% failure | |
|---|---|---|---|---|---|---|
| Agents | Success | Collisions | Success | Collisions | Success | Collisions |
| 8 | 1.00 ± 0.00 | 0.5 ± 0.2 | 1.00 ± 0.00 | 1.6 ± 0.6 | 1.00 ± 0.00 | 2.4 ± 0.8 |
| 16 | 1.00 ± 0.00 | 2.7 ± 0.5 | 1.00 ± 0.00 | 8.2 ± 1.4 | 1.00 ± 0.00 | 12.9 ± 2.2 |
| 24 | 1.00 ± 0.00 | 9.5 ± 2.0 | 0.99 ± 0.01 | 35.7 ± 7.3 | 0.98 ± 0.03 | 56.4 ± 12.5 |
| 32 | 1.00 ± 0.00 | 28.9 ± 8.4 | 0.95 ± 0.04 | 106.5 ± 25.7 | 0.94 ± 0.04 | 163.5 ± 32.3 |
| 48 | 0.98 ± 0.03 | 133.7 ± 41.2 | 0.81 ± 0.07 | 441.4 ± 96.4 | 0.61 ± 0.09 | 645.3 ± 90.2 |
| 64 | 0.92 ± 0.05 | 412.4 ± 101.9 | 0.51 ± 0.09 | 1099.5 ± 161.9 | 0.10 ± 0.05 | 1468.1 ± 157.5 |

Table 4: Communication failure test results on `Random` maps. Zero message failure means the original LC-MAPF performance. The provided values are average ± 95% confidence interval. Best results are marked with background color per row.

The success rates comparison demonstrates the negative effect of random noise messages compared to the LC-MAPF, proving the importance of the information communicated in LC-MAPF messages for achieving performance improvement. However, despite the extreme experimental setup with 50% message failure probability, the agents can successfully solve simple tasks with 8 and 16 agents, and obtain partial success in cases with larger agent populations.

## B COMMUNICATION BANDWIDTH

LC-MAPF enables communication between an agent and a fixed set of 12 local neighbors. This limit was chosen because a collision is possible only with an agent that occupies one of the 12 cells closest to the current location. Communication is not strictly restricted to agents within these cells; other agents within the observable area can still participate in communication. However, agents in these cells are prioritized due to their proximity.

For clarity, Figure 4 illustrates the relevant locations and actions that may result in a collision. The first four agents are positioned in the cardinally adjacent cells, and a collision with them is possible if both agents attempt to swap positions or if one agent chooses to wait. The next four potentially colliding agents are located in the diagonally adjacent cells; each of these has two possible actions that could lead to a collision, specifically in cases where both agents choose to enter the same cell. The final four agents are located two cells away, but a collision with them remains possible depending on the chosen actions. Regardless of the actions taken by agents in any other location, a collision with them in the current step is impossible.

We acknowledge that limited communication can affect performance. However, this constraint is motivated by practical considerations: in real-world applications, the bandwidth of communication channels is typically limited. Additionally, communication incurs costs; for instance, sending messages can significantly drain a robot's battery. Thus, there is an inherent trade-off between performance and communication bandwidth or cost.

On the one hand, in our approach, this limitation can be partially mitigated through chain communication. That is, agent $A$ may communicate with agent $B$ at communication round $t$, and agent $B$ may subsequently communicate with agent $C$ at round $t + 1$. As a result, information from agent $A$ can be propagated to agent $C$ (with some delay), even though $A$ and $C$ do not directly communicate.

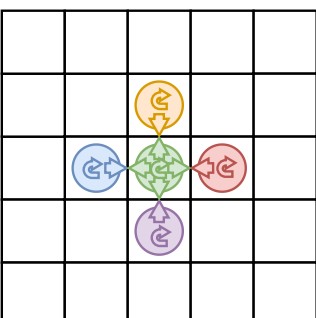 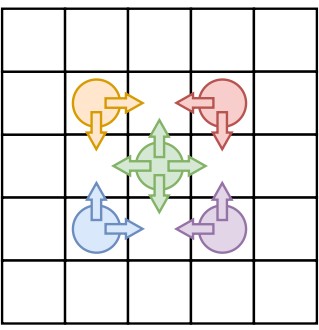 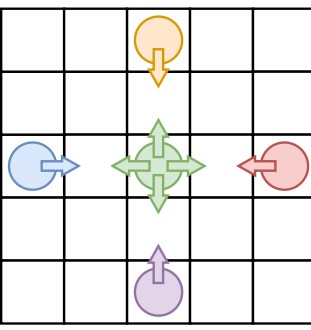

Figure 4: All agents and the corresponding actions that may result in a collision with the reference agent (marked as green).

| Agents | Limit = 1 | Limit = 2 | Limit = 4 | Limit = 8 | Limit = 13 |
|---|---|---|---|---|---|
| 8 | 0.99 ± 0.01 | 1.00 ± 0.00 | 1.00 ± 0.00 | 1.00 ± 0.00 | 1.00 ± 0.00 |
| 16 | 0.93 ± 0.04 | 1.00 ± 0.00 | 1.00 ± 0.00 | 1.00 ± 0.00 | 1.00 ± 0.00 |
| 24 | 0.80 ± 0.07 | 0.97 ± 0.03 | 1.00 ± 0.00 | 1.00 ± 0.00 | 1.00 ± 0.00 |
| 32 | 0.55 ± 0.09 | 0.90 ± 0.05 | 1.00 ± 0.00 | 0.99 ± 0.01 | 1.00 ± 0.00 |
| 48 | 0.16 ± 0.06 | 0.47 ± 0.09 | 0.88 ± 0.05 | 0.95 ± 0.04 | 0.98 ± 0.03 |
| 64 | 0.07 ± 0.04 | 0.20 ± 0.07 | 0.48 ± 0.09 | 0.73 ± 0.08 | 0.73 ± 0.08 |

Table 5: Success rates for different sizes of the communication neighborhood evaluated on `Mazes` maps. The Limit value shows the number of communicating agents. Limit = 13 refers to the original LC-MAPF. The provided values are average ± 95% confidence interval. Best results are marked with background color per row.

On the other hand, to prove the effectiveness and efficiency of the proposed neighborhood size, we test more restrictive neighborhood sizes on the Mazes maps from the POGEMA benchmark. We limit the number of communicating agents in LC-MAPF to 1, 2, 4, 8, and 13. For example, Limit = 2 means that each agent receives messages from at most 2 agents (including itself). The agents are sorted based on their distance to the current agent. Thus, when there are 5 agents in observation, an agent will receive only two messages from the closest ones. The same logic applies to the upper bound. If the actual number of agents present in the local field of view is greater than 13, only the closest 13 will be taken into account. The results are presented in Table 5.

The results clearly demonstrate that limiting the number of communicating agents significantly reduces the success rate, especially when the most strict limitations (4 agents or fewer) are applied to instances with larger agent populations (48 and 64 agents). The absence of significant influence from tighter limits on instances with fewer agents is explained by the fact that the actual number of agents present in the observations on such instances satisfies the reduced limit in most cases. This experiment highlights the importance of communication for LC-MAPF and demonstrates that limiting it can negatively impact performance.

Another dimension of LC-MAPF communication bandwidth is the size of the message vector that is used by the agents for information exchange. To test how the message vector size affects the LC-MAPF performance, we modified the message generation process so that the generated message vector is projected to a space with 4 times less dimensionality and then projected to the original size, as message vectors are required to have a size equal to the model hidden size to be processed correctly. The results of the experiment on Mazes maps are presented in Table 6.

Both success rate and number of collisions become worse after reducing the inner dimensionality of the message vector, which demonstrates the effectiveness of the proposed LC-MAPF configuration.

## C   LARGE-SCALE EVALUATION

| | Success Rate | | Number of Collisions | |
|---|---|---|---|---|
| Agents | Msg=160 | Msg=40 | Msg=160 | Msg=40 |
| 8 | 1.00 ± 0.00 | 1.00 ± 0.00 | 0.48 ± 0.16 | 0.60 ± 0.21 |
| 16 | 1.00 ± 0.00 | 1.00 ± 0.00 | 3.63 ± 0.66 | 4.84 ± 1.05 |
| 24 | 1.00 ± 0.00 | 1.00 ± 0.00 | 11.99 ± 1.57 | 12.91 ± 2.01 |
| 32 | 1.00 ± 0.00 | 1.00 ± 0.00 | 31.73 ± 4.11 | 35.08 ± 5.17 |
| 48 | 0.95 ± 0.04 | 0.90 ± 0.05 | 141.23 ± 22.22 | 183.42 ± 36.07 |
| 64 | 0.72 ± 0.07 | 0.71 ± 0.08 | 562.46 ± 97.01 | 638.62 ± 98.22 |

Table 6: Effect of the message vector size on LC-MAPF performance on `Mazes` maps. Each metric is reported for message size 160 (proposed) and 40 (reduced). Decreasing the message size negatively affects both success rate and number of collisions. The reported values are mean ± 95% confidence interval.

The main series of experiments was conducted on the POGEMA benchmark, where the maximum number of agents in the instances varies from 64 to 256 depending on the type of map. Although we have already demonstrated that LC-MAPF scales linearly with the number of agents, we also wanted to show that LC-MAPF can scale to thousands of agents and solve instances with such large numbers of agents. However, the exponential growth in runtime of other learning-based approaches with communication, such as SCRIMP and DCC, prevents us from making comparisons with them on large maps containing thousands of agents. We evaluated LC-MAPF on 256×256 random maps (with obstacle densities of 10% and 20%) and empty maps (0% density) with up to 5,000 agents. The results of this experiment are shown in Table 7. The reported numbers correspond to the

| | Density | | |
|---|---|---|---|
| Agents | 0% | 10% | 20% |
| 1000 | 448 | 462 | 456 |
| 2000 | 456 | 480 | 480 |
| 3000 | 471 | 494 | 516 |
| 4000 | 475 | 481 | 598 |
| 5000 | 504 | 614 | 1024 |

Table 7: Number of steps required to solve the corresponding instance depending on obstacle density and the number of agents in the instance.

makespan, i.e., the number of steps required for all agents to reach their goal locations (and occupy them simultaneously). When the value 1024 is reported, the corresponding instance was not successfully solved and was terminated. The scalability of LC-MAPF remained linear in this experiment – it requires approximately 0.5 seconds per step for instances with 1,000 agents and 2.5 seconds for instances with 5,000 agents.

## D    DETAILED RESULTS

In Figure 5 and Figure 6 we provide the detailed comparisons of LC-MAPF and baselines in success rates and sum-of-costs ratios, rspectively, for each map type used in experimental evaluation.

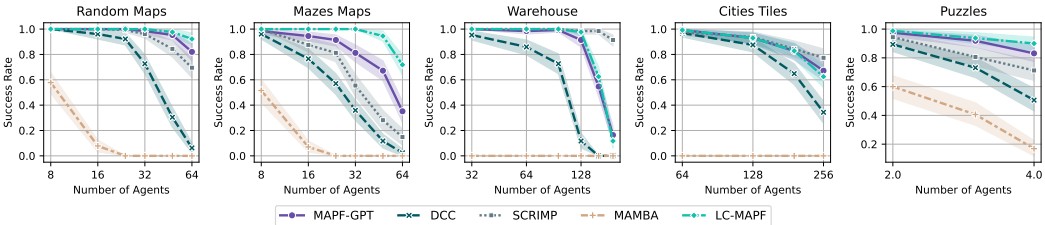

Figure 5: The detailed success rates of LC-MAPF and baselines for each map type. The shaded area indicates 95% confidence intervals.

Table 8 we list the aggregated (i.e. based on the all evaluated instances with 8-64 agents for Maze and 2-4 agents for Puzzles) results for the Maze and Puzzle environments support the Figure 3 and demonstrate the performance trade-offs against existing baselines.

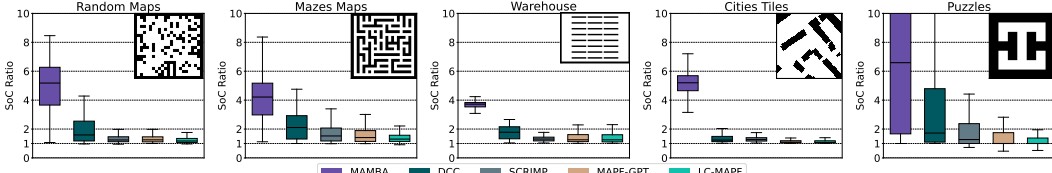

Figure 6: The comparison of LC-MAPF and baselines' sum-of-costs for each map configuration. Whiskers indicate 95% confidence intervals.

| Algorithm | Success Rate | SoC | Makespan | Collisions |
|---|---|---|---|---|
| **Mazes** | | | | |
| LC-MAPF-2M | 0.94 ± 0.02 | 1214.16 ± 94.86 | 53.40 ± 2.03 | 125.26 ± 23.41 |
| MAPF-GPT-2M | 0.78 ± 0.03 | 1452.79 ± 116.50 | 69.20 ± 2.69 | 242.24 ± 40.10 |
| SCRIMP | 0.61 ± 0.03 | 1519.17 ± 120.06 | 76.47 ± 3.09 | 0.0 |
| DCC | 0.47 ± 0.03 | 1994.17 ± 139.76 | 90.27 ± 3.18 | 84.82 ± 8.05 |
| MAMBA | 0.10 ± 0.02 | 3177.16 ± 178.27 | 119.92 ± 1.83 | 744.22 ± 64.43 |
| LaCAM* | 0.98 ± 0.01 | 767.11 ± 58.30 | 38.11 ± 1.33 | 0.0 |
| **Puzzles** | | | | |
| LC-MAPF-2M | 0.98 ± 0.01 | 41.10 ± 6.73 | 15.02 ± 2.07 | 2.80 ± 0.86 |
| MAPF-GPT-2M | 0.94 ± 0.02 | 63.31 ± 10.58 | 21.44 ± 3.07 | 7.73 ± 2.21 |
| SCRIMP | 0.85 ± 0.03 | 77.86 ± 11.20 | 30.29 ± 4.14 | 0.0 |
| DCC | 0.74 ± 0.04 | 93.50 ± 11.44 | 43.62 ± 4.92 | 1.99 ± 0.69 |
| MAMBA | 0.40 ± 0.04 | 173.60 ± 14.74 | 81.14 ± 5.41 | 29.61 ± 4.88 |
| LaCAM* | 1.00 ± 0.00 | 20.84 ± 1.63 | 8.38 ± 0.50 | 0.0 |

Table 8: The detailed results on success rates, sum-of-costs (SoC), makespan, and the number of collisions on `Maze` and `Puzzles` maps. LaCAM* and SCRIMP have zero collisions due to their usage of centralized solvers. The reported values are mean ± 95% confidence interval.

# E DYNAMIC OBSTACLES

In this section, we demonstrate the robustness and adaptability of LC-MAPF in the case of a dynamic obstacle configuration. We modified both the Random and Mazes environments by introducing the following stochastic dynamics. At each time step, every obstacle in the environment could be either removed or re-added with a probability of 0.05. To preserve feasibility and prevent deadlocks, we ensured that an agent's current cell is never converted into an obstacle. The resulting success rates and number of collisions are presented in Table 9.

| | **Random** | | **Mazes** | |
|---|---|---|---|---|
| Agents | Success Rate | Collisions | Success Rate | Collisions |
| 8 | 1.00 ± 0.00 | 0.89 ± 0.29 | 1.00 ± 0.00 | 1.10 ± 0.33 |
| 16 | 1.00 ± 0.00 | 4.76 ± 0.79 | 1.00 ± 0.00 | 5.07 ± 0.72 |
| 24 | 1.00 ± 0.00 | 12.57 ± 1.68 | 1.00 ± 0.00 | 14.16 ± 1.61 |
| 32 | 1.00 ± 0.00 | 27.26 ± 2.73 | 1.00 ± 0.00 | 30.39 ± 2.95 |
| 48 | 0.98 ± 0.03 | 88.73 ± 7.46 | 0.98 ± 0.02 | 100.82 ± 8.31 |
| 64 | 0.93 ± 0.04 | 216.58 ± 18.40 | 0.95 ± 0.04 | 235.39 ± 15.28 |

Table 9: Success rates and number of collisions for LC-MAPF in dynamic obstacles scenario on Random and Mazes maps. The reported values are mean ± 95% confidence interval.

For this experiment, we do not re-train LC-MAPF in dynamic settings and employ the dynamic scenario only for execution. LC-MAPF maintains high success rates with only minor degradation

as the number of agents increases. These results highlight the strengths of a learnable, decentralized approach.

To note, success rates in the Mazes environment with dynamic obstacles are marginally higher than in the static execution scenario reported in the main paper. This is because the occasional removal of obstacles can create shortcuts or open alternative paths, reducing congestion and making navigation easier in structured maze layouts.

## F    THE USE OF LARGE LANGUAGE MODELS (LLMS)

LLMs were used exclusively for text polishing and editing (e.g., grammar, spelling, word choice).

