# OpenReview forum: "Learning to Communicate Locally for Large-Scale Multi-Agent Pathfinding"
_ICLR.cc/2026/Conference — ICLR 2026 Conference Withdrawn Submission_

### Official Review · Reviewer_5HHG · 2025-10-31

**Soundness:** 1
**Presentation:** 2
**Contribution:** 2
**Rating:** 2
**Confidence:** 4

**Summary:**

The paper presents a learnable communication module for Multi-Agent Path Finding (MAPF). The communication module leverages a transformer-based approach with 2 million parameters, used to amplify a Decentralized MAPF approach trained via Imitation Learning of a centralized method. Experiments were conducted in the POGEMA environment, and support the hypothesis.

**Strengths:**

- Detailed explanation of the methodology

- The experiments show that the proposed method is able to outperform baselines in large scales.

- Extensive ablation studies show that multi-round communication improves performance.

**Weaknesses:**

- Limited literature review in MAPF problems, especially in MARL and Communication-based approaches.

- While the experimental results show that the method is able to outperform the two baselines in 192 agent scale, it is not able to achieve similar performance in smaller scales.

- Limited number of Baselines (2).

**Questions:**

- It is unclear how the Communication model is pretrained (as GPT implies Generative Pretrained Transformer).

- How does the proposed approach compare against other learning based communication methods in MARL settings presented in [1] such as [2]?

[1] M. Bettini, A. Shankar, and A. Prorok, “Heterogeneous Multi-Robot Reinforcement Learning,” Jan. 17, 2023, _arXiv_: arXiv:2301.07137. doi: [10.48550/arXiv.2301.07137](https://doi.org/10.48550/arXiv.2301.07137).

[2] E. Seraj _et al._, “Learning Efficient Diverse Communication for Cooperative Heterogeneous Teaming,” in _Proceedings of the 21st International Conference on Autonomous Agents and Multiagent Systems_, in AAMAS ’22. Richland, SC: International Foundation for Autonomous Agents and Multiagent Systems, 2022, pp. 1173–1182.

---

> ### Author Response · Authors · 2025-11-21
> **Author Response**
>
> We sincerely thank the reviewer for their detailed and constructive feedback. We greatly appreciate the positive comments on the paper’s detailed explanation of the methodology, experiments showing that LC-MAPF outperforms baselines on large scales, and extensive ablation studies.
>
> **W1:**
> Thank you for that comment. We were limited by the size of the article, but we are ready to expand the related works section and add the necessary works. We will be glad to receive your recommendations. For our part, we plan to expand it by discussing such works as:
>
> 1) “Where Paths Collide: A Comprehensive Survey of Classic and Learning-Based Multi-Agent Pathfinding” (Wang et al., 2025)
>
> 2) “Graph Attention Networks Based Multi-Agent Path Finding via Deep Reinforcement Learning” (Zhang et al., PLOS ONE, 2025)
>
> 3) “Multi-Agent Path Finding with Prioritized Communication Learning” (Li et al., ICRA 2022)
>
> 4) AC2C: Adaptively Controlled Two-Hop Communication for Multi-Agent Reinforcement Learning https://arxiv.org/abs/2302.12515
>
> 5) Dynamic Graph Communication for Decentralised Multi-Agent Reinforcement Learning https://www.alphaxiv.org/abs/2501.00165v1
>
> 6) Contextual Knowledge Sharing in Multi-Agent Reinforcement Learning with Decentralized Communication and Coordination https://arxiv.org/abs/2501.15695
>
> We are also committed (space permitting) to add recent classical MAPF approaches to the review:
>
> 7) Okumura, K., 2024, May. Engineering LaCAM*: Towards Real-time, Large-scale, and Near-optimal Multi-agent Pathfinding. In Proceedings of the 23rd International Conference on Autonomous Agents and Multiagent Systems (pp. 1501-1509).
>
> 8) Zang, H., Zhang, Y., Jiang, H., Chen, Z., Harabor, D., Stuckey, P.J. and Li, J., 2025, April. Online guidance graph optimization for lifelong multi-agent path finding. In Proceedings of the AAAI Conference on Artificial Intelligence (Vol. 39, No. 14, pp. 14726-14735).
>
> 9) Veerapaneni, R., Saleem, M.S., Li, J. and Likhachev, M., 2025, April. Windowed MAPF with Completeness Guarantees. In Proceedings of the AAAI Conference on Artificial Intelligence (Vol. 39, No. 22, pp. 23323-23332).
>
> **W2:**
> Regarding decision time, presented in Table 3: unlike DCC and SCRIMP, LC-MAPF applies a multi-round communication procedure that allows agents to iteratively improve their intermediate decisions before finalizing them. In cases of relatively small agent populations (128 agents or fewer), this iterative process results in higher decision time with respect to the baselines, while in the large-scale environment with 192 agents, LC-MAPF shows the minimal decision time.
>
> Regarding the general performance of the method, in Figure 5 of Appendix D of the updated version of the paper, we provide the detailed comparison of LC-MAPF against the baselines, including DCC and SCRIMP, in terms of success rates. In scenarios with up to 128 agents, LC-MAPF shows the top performance compared to DCC and SCRIMP.
>
> **W3:**
> For our baselines, we selected the four methods that are best-performing on POGEMA benchmark [1] to solidify the experimental assessment of LC-MAPF in addition to the main MAPF-GPT baseline. In this study, we focus on the imitation learning paradigm, following the POGEMA benchmark evaluations that empirically showed that evaluated imitation learning approaches (DCC, SCRIMP) outperform the RL-based ones (IQL, QMIX,VDN, and QPLEX).
>
>
> **Q2:**
> LC-MAPF is an imitation learning-based approach for MAPF. To ensure the fairest comparison and alignment of the proposed approach with related works, we consider baseline methods that are primarily focused on MAPF and employ the imitation learning paradigm.
>
>
> Both suggested methods are RL-based training and specifically address the cooperative heterogeneity challenges, providing the heterogeneity-tailored communication mechanisms tailored and introducing specific training paradigms to tackle the composite team policies. The objectives of these works are fundamentally different from the one presented in our paper. Including such baselines would be irrelevant because it doesn't address the main MAPF challenges e.g. collision avoidance or path coordination in a shared environment and can mislead the reader.
>
> **Q1:** We adopted the GPT model architecture as a backbone for the LC-MAPF architecture. Our model with communication was trained from scratch.
>
> [1] Skrynnik, Alexey, et al. "POGEMA: A Benchmark Platform for Cooperative Multi-Agent Pathfinding." The Thirteenth International Conference on Learning Representations, 2025.

---

### Official Review · Reviewer_aoCS · 2025-11-01

**Soundness:** 2
**Presentation:** 2
**Contribution:** 2
**Rating:** 4
**Confidence:** 4

**Summary:**

This paper presents LC-MAPF, a novel decentralized learning framework for multi-agent pathfinding (MAPF) that incorporates learnable local communication among agents. The proposed approach builds upon imitation learning, training agents on expert demonstrations from centralized solvers (LaCAM*), but introduces a multi-round message-passing mechanism that allows agents to exchange information with nearby peers without explicit communication supervision.

**Strengths:**

- The paper introduces a learnable communication framework for decentralized MAPF without direct supervision on communication messages—an elegant and conceptually novel design. The multi-round local communication implemented via a transformer resembles end-to-end differentiable message passing, bridging imitation learning and graph-based communication paradigms.
- The methodology is solid and well-grounded in both MAPF and multi-agent learning literature. The derivation of gradient flow for communication learning (Eq. 10–11) is mathematically clear and demonstrates how communication content is implicitly optimized for coordination. The ablation and scalability analyses provide convincing empirical support.

**Weaknesses:**

- The model uses a fixed number of communication rounds and neighbors, which may not optimally adapt to different map densities or task complexities. Adaptive mechanisms for determining when and with whom to communicate could further enhance efficiency.
- Although the paper compares to leading IL-based solvers, it would strengthen the contribution to also benchmark against communication-learned RL frameworks (e.g., CommNet, DIAL, or QMIX with communication extensions).
- Although LC-MAPF scales linearly, its per-step latency is higher for smaller agent populations. A discussion of optimization or deployment strategies (e.g., asynchronous communication, sparse attention) would improve the practical relevance.

**Questions:**

- Have the authors considered dynamic adjustment of the number of communication rounds based on environment complexity or agent density? This could improve efficiency without sacrificing coordination.
- Since the communication vectors are learned implicitly, have the authors attempted to visualize or interpret what information the messages encode (e.g., goals, conflict zones, local intents)?
- The method assumes homogeneous agents. How difficult would it be to extend LC-MAPF to heterogeneous agent systems with different capabilities or goals?

---

> ### Author Response · Authors · 2025-11-21
> **Author Response**
>
> We thank the reviewer for their constructive feedback. We appreciate the acknowledgment of LC-MAPF’s elegant and conceptually novel design, bridging imitation learning and graph-based communication paradigms, solid and well-grounded methodology, and convincing ablation and scalability analyses.
>
> **W1 & Q1:**
> Thank you for your suggestion. Let us firstly emphasize that the LC-MAPF number of communication neighbors is, in fact, adaptive, as each agent communicates only with nearest neighbors within its local field of view, capped at 13 agents in our approach. In Appendix B Figure 4, we provide the results of ablation on the number of communicating agents. It shows the performance decrease for reducing the number of communication partners.
>
> Dynamically adjusting communication rounds presents significant technical challenges. A key difficulty is that even when an agent stops communicating, it can still be indirectly affected by messages exchanged between its neighbors. This interdependence makes it difficult to reduce communication in a principled manner. To maintain consistency and tractability in our experiments, we therefore adopted a fixed communication scheme. However, developing adaptive communication strategies remains a promising avenue for future work.
>
> **W2:**
> As our baselines, we selected the methods best-performing on POGEMA benchmark [1] to solidify the experimental assessment of LC-MAPF. In this study, we focus on the imitation learning paradigm.
>
> [1] Skrynnik, Alexey, et al. "POGEMA: A Benchmark Platform for Cooperative Multi-Agent Pathfinding." The Thirteenth International Conference on Learning Representations, 2025.
>
> **W3:** Thank you for your suggestion. To mitigate the described negative effect on latency, for scenarios with smaller number of agents, the number of communication rounds can be decreased down to 2 rounds (or less, depending on the actual size of the considered agent population) without performance loss (as shown in Table 2 of the main paper).
>
> We have added the corresponding discussion at the end of the Section 5.2 of the revised version of the paper.
>
> **Q2:**
> Thank you for your question. The visualization of messages is indeed an interesting future direction. However, it is out of the scope of the current study.
>
> **Q3:**
> The implementation of the LC-MAPF architecture does not rely on the agents’ homogeneity assumption, assuming the seamless extension to heterogeneous systems. The communication procedure is performed in the hidden space of the model and does not depend on agent capabilities or goals, as soon as their initial presentations can be encoded into the hidden space of the model.

---

> > ### Comment · Reviewer_aoCS · 2025-11-28
> >
> > Thank you for the reply and clarifications. The authors have answered some of my main concerns and I will raise my score.

---

> > > ### Author Response · Authors · 2025-11-28
> > >
> > > Thank you so much for your thoughtful engagement with our work and for raising your score.
> > >
> > > Are there any remaining points we could clarify or expand upon to fully address your concerns?

---

### Official Review · Reviewer_CJ6q · 2025-11-03

**Soundness:** 3
**Presentation:** 3
**Contribution:** 3
**Rating:** 8
**Confidence:** 3

**Summary:**

This paper introduces a new learning-based MAPF method (LC-MAPF) that focuses on improving coordination through agents' communication. The model employs a transformer-based backbone, similar to prior work such as MAPF-GPT, but extends it by concatenating message embeddings from neighboring agents along with local observations as model input. This is a simple yet effective idea, supported by the results on the POGEMA benchmark.

**Strengths:**

- The contribution is well-executed (albeit coming across as a bit incremental)

**Weaknesses:**

1. The authors describe LC-MAPF as a foundation model, but this term is reserved for architectures trained on large-scale, diverse datasets that generalize across multiple tasks. Since LC-MAPF is trained exclusively on MAPF data and evaluated only on MAPF environments, this characterization seems overstated.

2. Theoretical novelty is limited as the backbone architecture, data collection procedure, and training pipeline closely mirror MAPF-GPT. The main innovation lies in incorporating message passing, which, although useful, is a relatively incremental extension.

3. While Figure-3 and the Appendix provide partial insights, it would be valuable to include a comprehensive comparison table for the Maze and Puzzle environments summarizing success rate, makespan, and collision rate. This would help readers clearly assess performance trade-offs against existing baselines.

**Questions:**

1) Could you clarify how this paper represents a leap over previous work?
2) Could you clarify whether for training, a foundation model was used or was it just MAPF data?

---

> ### Author Response · Authors · 2025-11-21
> **Author Response, part 1.**
>
> We thank the reviewer for their detailed and insightful feedback. We appreciate the recognition of the well-executed contribution and the effective idea of LC-MAPF.
>
> **W1 & Q2:**
> We follow the well-established definition of the foundation model as a neural network model trained on a large array of disparate data and demonstrating its ability to perform zero-shot/few-shot learning to perform new tasks. In our work, LC-MAPF is really trained on a large dataset, which, from the point of view of the MAPF task, is varied in terms of conditions and maps. For MAPF tasks, the common approach is to tune the method for each map separately (e.g., SILLM[1]). In contrast, we train LC-MAPF on a large mixture of Mazes and Random maps and evaluate it using a wide variety of maps and agent populations unseen during training. Moreover, the MAPF task is naturally susceptible to even slight changes in the environment or agent population size that can lead to drastic shifts in the optimal strategy. LC-MAPF aims to optimize the model for a single, general policy rather than create per-map specialists. It is also shown that models similar to ours show zero-shot learning abilities (see MAPF-GPT). Following this paradigm, we describe LC-MAPF as a foundation model.
>
>
> [1] Jiang, He, et al. "Deploying ten thousand robots: Scalable imitation learning for lifelong multi-agent path finding." 2025 IEEE International Conference on Robotics and Automation (ICRA). IEEE, 2025.

---

> > ### Author Response · Authors · 2025-11-21
> > **Author Response, part 2.**
> >
> > **W2 & Q1:**
> > MAPF is a constantly evolving field, and it is not uncommon that adding a single novel enhancement to a well-established MAPF method boosts its performance and pushes the limit of MAPF in general. For example, there exists a plethora of papers dedicated to adding heuristics [1], prioritization schemes [2], and other heuristic improvements [3] to the well-established CBS [4] algorithm.
> >
> > In our paper, we indeed follow one of the most recent and effective approaches to tackle MAPF - massive pre-training on expert data, MAPF-GPT. Regarding the related works on imitation learning on expert data, like DAGGER [5,6], which applies expert access to augment training data, LC-MAPF learns multi-agent communication without supervision on messages; only action demonstrations are provided. The message content and interpretation are *learned* by the agents.
> >
> > On top of that, we add learnable communication into the pipeline. LC-MAPF differs from classic approaches like GNNs [7,8] in the following. Instead of communicating with all adjacent nodes as in standard GNNs, LC-MAPF dynamically selects neighbors based on each agent’s egocentric locality, effectively pruning the communication graph to nearby agents and making the topology dynamic rather than static. Additionally, LC-MAPF uses a Transformer to aggregate incoming messages alongside the current observation, enabling flexible and learned integration. This contrasts with typical GCNs [9], which use fixed mean or sum aggregations, and GATs [8], which apply fixed attention; here, each agent processes local messages and decides how to update its own message for the next round.
> >
> > Moreover, LC-MAPF presents a multi-round message passing, which is suggested for the first time in the literature, to the best of our knowledge. The related communicative methods like PRIMAL [10], DHC [11], DCC [12], and SCRIMP [13], and general multi-agent communication approaches [14-18] present only single-pass communication mechanisms.
> >
> >
> > As a result, LC-MAPF exhibits stronger performance and outperforms the state of the art.
> > Practically, the approach scales decision time linearly with the number of agents, resolving the scalability bottleneck that is common for previous communication-based MAPF solvers like DCC and SCRIMP.
> >
> >
> > [1] Felner, A., Li, J., Boyarski, E., Ma, H., Cohen, L., Kumar, T.S. and Koenig, S., 2018, June. Adding heuristics to conflict-based search for multi-agent path finding. In Proceedings of the International Conference on Automated Planning and Scheduling (Vol. 28, pp. 83-87)
> >
> > [2] Andreychuk, A., Yakovlev, K., Boyarski, E. and Stern, R., 2021, May. Improving continuous-time conflict based search. In Proceedings of the AAAI Conference on Artificial Intelligence (Vol. 35, No. 13, pp. 11220-11227)
> >
> > [3] Li, J., Harabor, D., Stuckey, P.J., Felner, A., Ma, H. and Koenig, S., 2019, July. Disjoint splitting for multi-agent path finding with conflict-based search. In Proceedings of the international conference on automated planning and scheduling (Vol. 29, pp. 279-283)
> >
> > [4] Sharon, G., Stern, R., Felner, A. and Sturtevant, N.R., 2015. Conflict-based search for optimal multi-agent pathfinding. Artificial intelligence, 219, pp.40-66.
> >
> > [5] Ross et al., "A Reduction of Imitation Learning and Structured Prediction to No-Regret Online Learning", AISTATS 2011
> >
> > [6] Anthony et al., "Thinking Fast and Slow with Deep Learning and Tree Search", NeurIPS 2017
> >
> > [7] Kipf et al., "Semi-Supervised Classification with Graph Convolutional Networks", ICLR 2017
> >
> > [8] Veličković et al., "Graph Attention Networks", ICLR 2018
> >
> > [9] Kipf, T. N. "Semi-supervised classification with graph convolutional networks." arXiv preprint arXiv:1609.02907 2016
> >
> > [10] Sartoretti, Guillaume, et al. "Primal: Pathfinding via reinforcement and imitation multi-agent learning." IEEE Robotics and Automation Letters 4.3 (2019): 2378-2385.
> >
> > [11] Ma, Ziyuan, Yudong Luo, and Hang Ma. "Distributed heuristic multi-agent path finding with communication." 2021 IEEE International Conference on Robotics and Automation (ICRA). IEEE, 2021.
> >
> > [12] Ma, Ziyuan, Yudong Luo, and Jia Pan. "Learning selective communication for multi-agent path finding." IEEE Robotics and Automation Letters 7.2 (2021): 1455-1462.
> >
> > [13] Wang, Yutong, et al. "Scrimp: Scalable communication for reinforcement-and imitation-learning-based multi-agent pathfinding." 2023 IEEE/RSJ International Conference on Intelligent Robots and Systems (IROS). IEEE, 2023.
> >
> > [14] Foerster et al., "Learning to Communicate with Deep Multi-Agent Reinforcement Learning", NeurIPS 2016
> >
> > [15] Cao et al., "Emergent Communication through Negotiation", ICLR 2018
> >
> > [16] Lowe et al., "On the Pitfalls of Measuring Emergent Communication", AAMAS 2019
> >
> > [17] Lowe et al., "On the Interaction between Supervision and Self-Play in Emergent Communication", ICLR 2020
> >
> > [18] Böhmer et al., "Deep Coordination Graphs", ICML 2020

---

> > > ### Author Response · Authors · 2025-11-21
> > > **Author Response, part 3.**
> > >
> > > **W3:**
> > > Below we provide the comparison tables for the Maze and Puzzle environments. Note that they are built aggregated, i.e., based on all evaluated instances with 8-64 agents for Maze and 2-4 agents for Puzzles. Also note that LaCAM has no collisions, as it’s a centralized search-based solver, and SCRIMP has no collisions as well, due to the integrated centralized collision-resolution technique, which is a mandatory part of this approach.
> > > The following table demonstrates the results on Mazes:
> > >
> > > | Algorithm           | Success Rate | SoC                       | Makespan      | Collisions          |
> > > | -           | - | -                    | -      | -   |
> > > | LC-MAPF-2M    | 0.94 ± 0.02      | 1214.16 ± 94.86   | 53.40 ± 2.03   | 125.26 ± 23.41 |
> > > | MAPF-GPT-2M | 0.78 ± 0.03      | 1452.79 ± 116.50  | 69.20 ± 2.69   | 242.24 ± 40.10 |
> > > | SCRIMP            | 0.61 ± 0.03      | 1519.17 ± 120.06 | 76.47 ± 3.09   | 0.0                    |
> > > | DCC                  | 0.47 ± 0.03      | 1994.17 ± 139.76 | 90.27 ± 3.18   | 84.82 ± 8.05     |
> > > | MAMBA             | 0.10 ± 0.02      | 3177.16 ± 178.27 | 119.92 ± 1.83 | 744.22 ± 64.43 |
> > > | LaCAM*             | 0.98 ± 0.01      | 767.11 ± 58.30     | 38.11 ± 1.33   | 0.0                    |
> > >
> > > Here is the table for Puzzles:
> > >
> > > | Algorithm           | Success Rate | SoC                    | Makespan      | Collisions   |
> > > | -           | - | -                    | -      | -   |
> > > | LC-MAPF-2M    | 0.98 ± 0.01      | 41.10 ± 6.73      | 15.02 ± 2.07   | 2.80 ± 0.86 |
> > > | MAPF-GPT-2M | 0.94 ± 0.02      | 63.31 ± 10.58    | 21.44 ± 3.07   | 7.73 ± 2.21 |
> > > | SCRIMP            | 0.85 ± 0.03      | 77.86 ± 11.20    | 30.29 ± 4.14   | 0.0              |
> > > | DCC                  | 0.74 ± 0.04      | 93.50 ± 11.44    | 43.62 ± 4.92   | 1.99 ± 0.69 |
> > > | MAMBA             | 0.40 ± 0.04      | 173.60 ± 14.74 | 81.14 ± 5.41 | 29.61 ± 4.88 |
> > > | LaCAM*             | 1.00 ± 0.00      | 20.84 ± 1.63     | 8.38 ± 0.50   | 0.0                |
> > >
> > > As you can see, in terms of success rate and solution cost (sum of cost and makespan), LC-MAPF significantly outperforms all learning- based approaches. In terms of the number of collisions, the proposed approach is second-best, while the best results demonstrate DCC, which also incorporates a communication mechanism.
> > >
> > > We have included these detailed results in Table 9 of Appendix D of the updated version of the paper to better demonstrate the advantages of the proposed method.

---

### Official Review · Reviewer_dsnY · 2025-11-10

**Soundness:** 3
**Presentation:** 3
**Contribution:** 2
**Rating:** 4
**Confidence:** 3

**Summary:**

Then paper introduces LC-MAPF, which is a decentralized framework for MAPF that is designed to improve large-scale multi-agent coordination through local communication mechanism. In this framework, each agent generate messages and exchanges them with nearby agents. Such communication proceeds in sequential rounds. The messages are learned end-to-end. The authors collect 32M samples and train LC-MAPF with 4 rounds of communication. Evaluation on the POGEMA dataset shows state-of-the-art performance among other decentralized MAPF solvers.

**Strengths:**

1. It is a novelty to explore end-to-end learned communication in MAPF. The authors adapt a multi-round comminucation in MAPF settings.
2. Experimental evaluation is comprehensive and strong in comparison agains baselines.

**Weaknesses:**

1. There is a lack of interpretability. The messages are learned end-to-end, but there is no analysis of what information is actually encoded.
2. Data collection is skewed towards maze-like maps, but without good justification. What is the motivation for this? Would it cause overfitting to the maze maps? Could you break down the results by map types?

**Questions:**

1. What is $L$ on line 186?
2. You provide the code but not the model weights. Do you plan to open source them as well?
3. On line 318, is it 15 billion of 1.5 billion?
4. Could you break down the results by map types?

---

> ### Author Response · Authors · 2025-11-21
>
> We thank the reviewer for their constructive feedback. We appreciate the recognition of the novelty of exploring end-to-end learned communication in MAPF and comprehensive evaluation and strong baseline comparison.
>
> **W1:**  We agree that understanding what is communicated is important, but see this as complementary to our main focus on designing and validating a coordination mechanism that scales to long horizons and many agents. Our contribution is to show, via ablations, that the proposed communication mechanism is crucial for performance and generalization; this is an architectural, not interpretability, claim.
>
> **W2:**
> The training dataset consists primarily of maze-like maps, with a small portion (10%) made up of random maps. This choice was motivated by two factors:
>  a) Maze-like maps are more challenging; if the model can efficiently solve instances on maze-like maps, it can handle random maps as well. The reverse, however, does not hold. We made some preliminary tests, trying to train a model with 50/50 proportions of maze-like and random maps and the results were substantially worse.
>  b) Both random and maze-like maps can be procedurally generated, unlike warehouse or city maps.
> Moreover, the main baseline approach, i.e. MAPF-GPT, uses the same dataset construction scheme, and we followed this paradigm to ensure a fair comparison.
>
> **Q1:**
> L stands for the length of the trajectory. We added this clarification in the updated version of the paper.
>
> **Q2:**
> The model weights are available in the weights folder of our paper repository: https://anonymous.4open.science/r/LC-MAPF-18734
>
> They were already included in the anonymous repository from the very first submission of the paper.
> We are committed to open-sourcing all related artifacts under permissive licenses, including the model weights.
>
>
> **Q3:** It is 1.5 billion. We fixed this issue in the updated version of the paper.
>
> **Q4:**
> We have added the per-map success rates and sum-of-costs ratios reports in Appendix D Figures 5 and 6 of the updated version of the paper.

---

### Comment · Area_Chair_xn6S · 2025-11-22

Dear Reviewers,

The authors have responded to your reviews. Please review and respond to their comments.

Best,
Your AC

---

### Note · Authors · 2025-12-02

**Comment:**

Dear Area Chair and Reviewers,

We sincerely appreciate the time and effort invested in reviewing our submission, as well as the thoughtful feedback provided. We are particularly grateful to the reviewer who expressed willingness to reconsider their assessment based on our comments. Your suggestions have helped us substantially improve the paper. Unfortunately, the leak on OpenReview prevented proper discussion during the review process.

In parallel with the submission, we independently made substantial revisions to the methodology, replacing the decoder-only architecture with a representation encoder and message decoder, decoupling message size from observation size, and integrating a modern, stabilizing Transformer stack. While the core idea of local multi-round communication remains, these updates significantly improve performance.
Given the extent of these changes, we believe the work merits a full review cycle rather than a mid-process adjustment.

We are deeply grateful to the Reviewers and Area Chair for their engagement and oversight. In light of these considerations, we have decided to withdraw the paper at this time.

**Withdrawal Confirmation:**

I have read and agree with the venue's withdrawal policy on behalf of myself and my co-authors.